# Inequality of the crowding-out effect of tobacco expenditure in Colombia

[Juan Miguel Gallego[1], Guillermo Paraje[1], Paul Rodríguez-Lesmes[2] *

1 School of Economics, Universidad del Rosario, Calle, Bogotá, Colombia, 2 Escuela de Negocios, Universidad Adolfo Ibáñez, Diagonal las Torres, Peñalolén, Santiago, Chile

* paul.rodriguez@urosario.edu.co

**Data Availability Statement:** Data is available at DANE's webpage. All codes required for reproduction and final derived datasets are available at: https://github.com/androdri1/tobacco_ineq.

## Abstract

In recent decades, policy initiatives involving increases in the tobacco tax have increased pressure on budget allocations in poor households. In this study, we examine this issue in the context of the expansion of the social welfare state that has taken place over the last two decades in several emerging economies. This study explores the case of Colombia between 1997 and 2011. In this period, the budget share of the poorest expenditure quintile devoted to tobacco products of smokers' households doubled. We analyse the differences between the poorest and richest quintiles concerning the changes in budget shares, fixing a reference population over time to avoid demographic composition confounders. We find no evidence of crowding-out of education or healthcare expenditures. This is likely to be the result of free universal access to health insurance and basic education for the poor. For higher-income households, tobacco crowds out expenditures on entertainment, leisure activities, and luxury expenditures. This finding should reassure policymakers who are keen to impose tobacco taxes as an element of their public health policy.

## 1. Introduction

The tobacco epidemic disproportionally affects low socioeconomic status (SES) households [1–6]. Global efforts such as the *World Health Organization Framework Convention on Tobacco Control* (WHO FCTC) have decades of promoting policies aimed at reducing smoking prevalence, especially tax increases, which have proved to be effective [7–12]. Yet, evidence shows that the demand for cigarettes is inelastic and households try to sustain their habits even if they have to reallocate their expenditures [13–15]. For low SES households that have continued to smoke, an important concern is the implications of the greater fiscal burden that they face as a result of tax modifications [11,16,17].

To compensate for the increase in prices, households need to decide where to draw resources from, resulting in a reduced standard of living [18,19]. One option available for households is to use income that they previously devoted to education and health expenditures. There is evidence of this crowding-out effect that impacts human capital accumulation (food intake, education, and health) and productive household investment in Bangladesh, rural China, Costa Rica, Ghana, India, Indonesia, Kenya, Malawi, Montenegro, Pakistan,

**Funding:** This project was funded under the GADC project by the CIHR/IDRC [grant number 108442-001], and Fulbright-Colciencias and Colombia Cientifica – Alianza EFI 60185 contract FP44842-220-2018, funded by The World Bank through the Scientific Ecosystems, managed by the Colombian Ministry of Science, Technology and Innovation (MINCIENCIAS). Guillermo Paraje acknowledges funding by ANID FONDECYT, 1201452.

**Competing interests:** The authors have declared that no competing interests exist.

Serbia, South Africa, Turkey, Vietnam, and Zambia [20–34]. Yet, in middle-income countries, an alternative is to use the resources that are saved by the expansion of social policy efforts. These policies involve transfers, that can be in cash or in kind, that provide an additional income to households that they might devote to the consumption of temptation goods. Evidence on the magnitude of this reallocation from the evaluation of cash-transfers programs is inconclusive [35].

In this paper, we explore the progression of the crowding-out effect of tobacco, along the income distribution, in the context of growing tobacco prices and social security expansion.

We consider the case of Colombia, a middle-income Latin American country. We analyse the changes in household budgets, across the income distribution, for smokers in comparison with non-smokers with similar observed characteristics. We use a repeated cross-section of the Colombian Quality of Life Survey (ECV, the acronym in Spanish) from 1997 to 2011. It includes household expenditure data during a period that saw increasing tobacco prices because of tighter tobacco control policies, and as a result, there was an increase in financial pressure on the lowest SES households of smokers. Alongside, access to health insurance and basic education increased notoriously in the study period and drastically reduced out-of-pocket expenditures in those areas. We present an overview of these two characteristics below in section 2.

To determine differences overtime on how budget-pressure of smoking affects the poorest households, we undertake two empirical steps. First, to establish a comparable group of smoking and non-smoking households each year, we use a genetic matching algorithm. Second, we contrast budget shares differences over total expenditures quintiles, between smokers and non-smokers. Alongside describing the dataset, section 3 presents the matching strategy of the empirical step 1. It also presents the statistical model required to obtain the estimates described in step 2. Results are presented in section 4, and section 5 presents the discussion and conclusions.

## 2. Context

### 2.1 Tobacco control policies

In an attempt to curb the tobacco epidemic, Colombia implemented a diverse range of control mechanisms that played an important role in the decrease in cigarette consumption. As part of the adoption of the WHO FCTC, in 2009, an anti-tobacco law (Law 1335 of 2009) was introduced that restricted smokers from consuming cigarettes in public areas. Then, in 2011, the government implemented the marketing restrictions included in Law 1335 of 2009. In addition, several tax-based reforms were introduced between 1997 and 2011. Since 1995, several low-powered tax increases have taken place involving specific contributions to sports, custom tariffs, and other consumption taxes. A major reform took place in 2010 (Law 1393 of 2010) when a unique tax was applied uniformly to both local and imported products, involving a combination of a lump-sum tax and an excise tax. During the study period, there was an increase of nearly 60% in the real average price per cigarette over the study period, as shown in Fig 1.

The policy initiatives outlined above were associated with a substantial reduction in the prevalence of smoking. In 1997, 25% of households reported consuming tobacco during the previous week, while this figure was down to around 10% by 2011. Concerning SES differentials, Panel A of Fig 2 shows prevalence by total expenditure quintiles from 1997 to 2011. In the initial year, while prevalence is larger for the first quintile to the fifth. By 2011, there is almost no difference across quintiles. This is in line with several studies that have found that initial differences in smoking prevalence across different characteristics have narrowed over

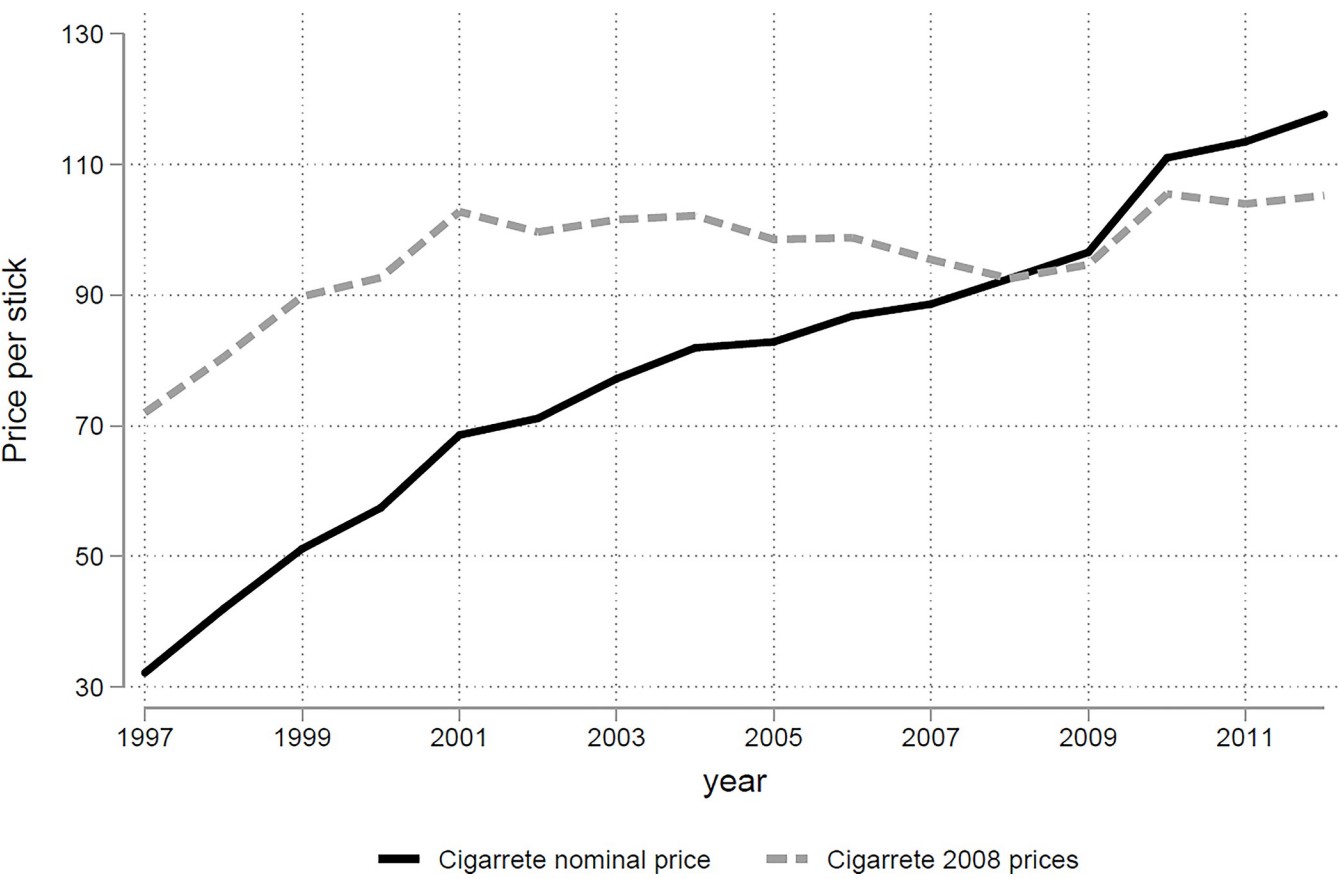

**Fig 1. Tobacco price evolution in Colombia.** Label: Authors' calculations based on Consumer Price Index data.

time [36–39]. Concerning expenditures, Panel B of Fig 2 presents a notorious SES gradient observed for the budget share allocated to tobacco by smoking households: while it remained the same for the richest quintile (less than 1%), it doubled for the poorest quintile, jumping from 3.1% to 6.2%. These average budget allocations are similar to the range of average international allocations, which vary from 1% in Mexico and Hong Kong to 10% in Zimbabwe and China [22].

## 2.2 Social policies

From 1997 to 2011 in Colombia, incomes grew and there was an important decline in poverty levels, for instance, extreme poverty fell from 16.9% to 6.6% [40]. This is a period in which Colombia, like other middle-income countries, introduced policies to reduce poverty and inequality, which might compensate for the potential financial pressures of tobacco tax increases. In particular, Colombia introduced a range of aggressive social policies aimed at reducing poverty such as universal health insurance and basic education. One of the most relevant improvements in the context of this study was the expansion of health insurance. Fig 3 shows health insurance coverage and self-reported health for smokers and non-smokers in quintiles 1 and 5 in the sample selected by the analysis (see below). These figures reflect the dramatic improvement in access to health insurance [41]. In 1997, approximately 80% of people in quintile 5 had insurance, but the figure was only 50% in quintile 1. In contrast, by 2011, nearly 90% of people had insurance regardless of their SES. As a result, Colombia has the second-

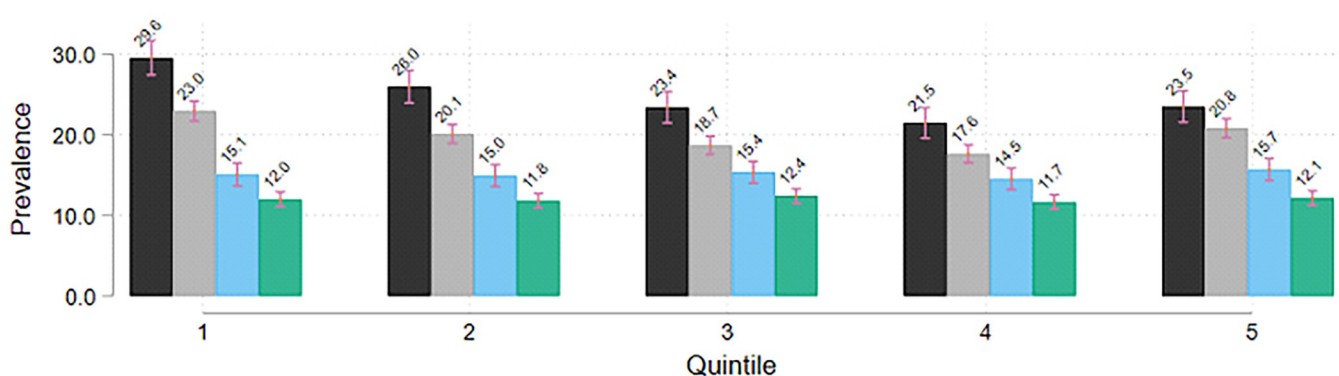

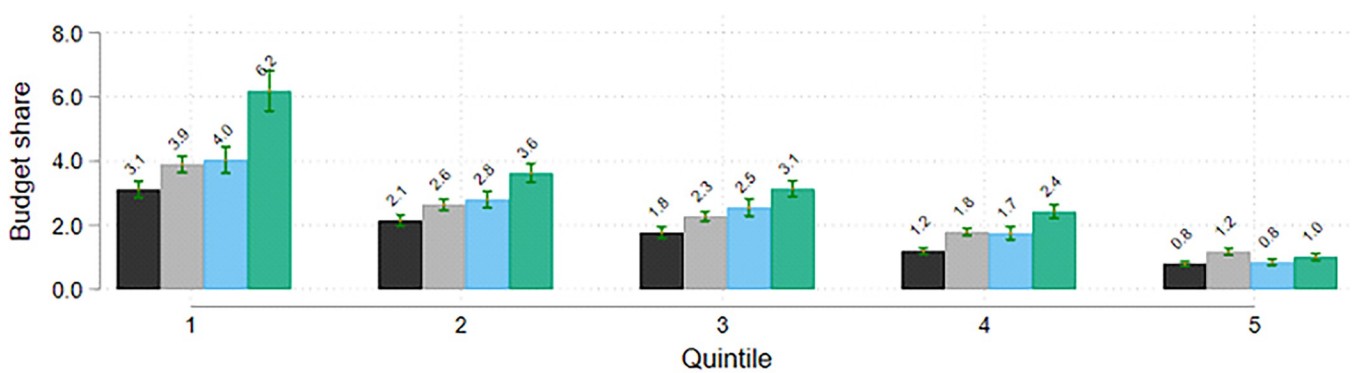

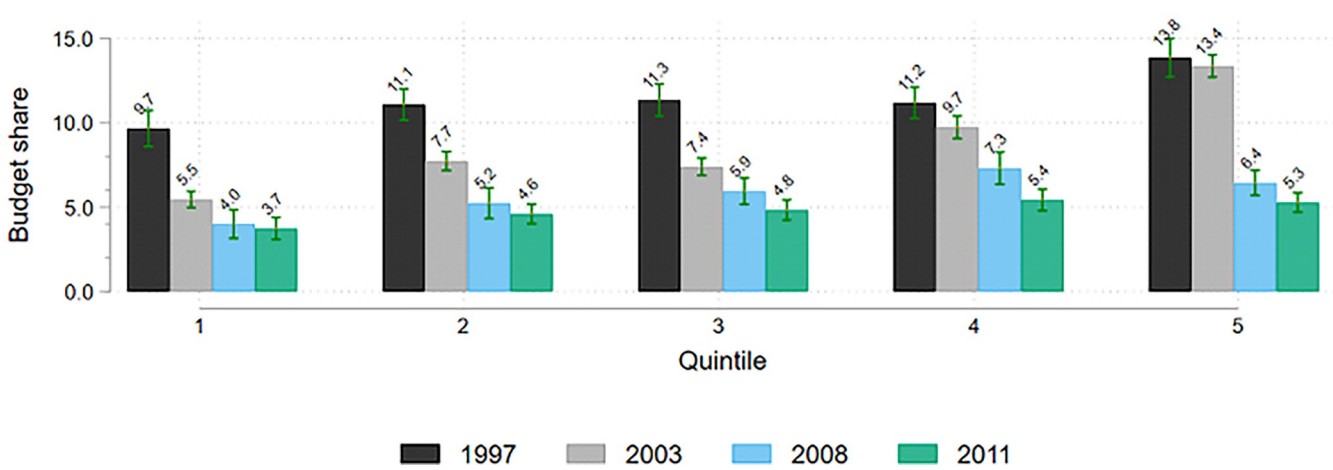

**Fig 2. Smoking prevalence and tobacco budget share by total expenditures.** Label: Authors' calculations.

lowest out-of-pocket health expenditures in Latin America [42]. A similar scenario is observed for education, and in our data, we observe a drastic reduction in both education and health expenditures for all households along the income distribution, as shown in Panel C of Fig 2.

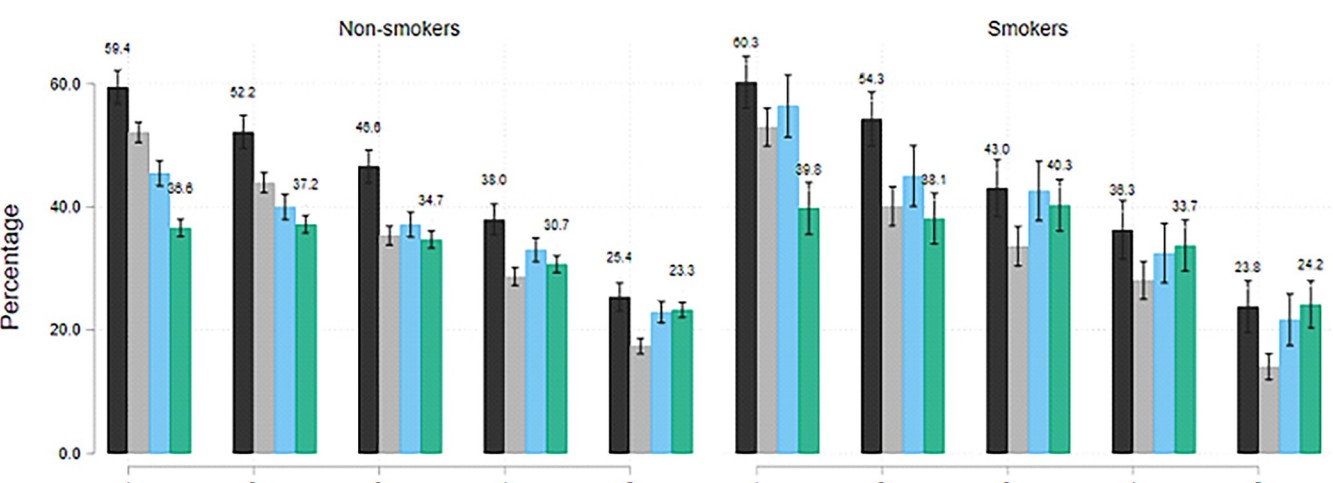

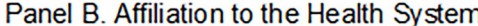

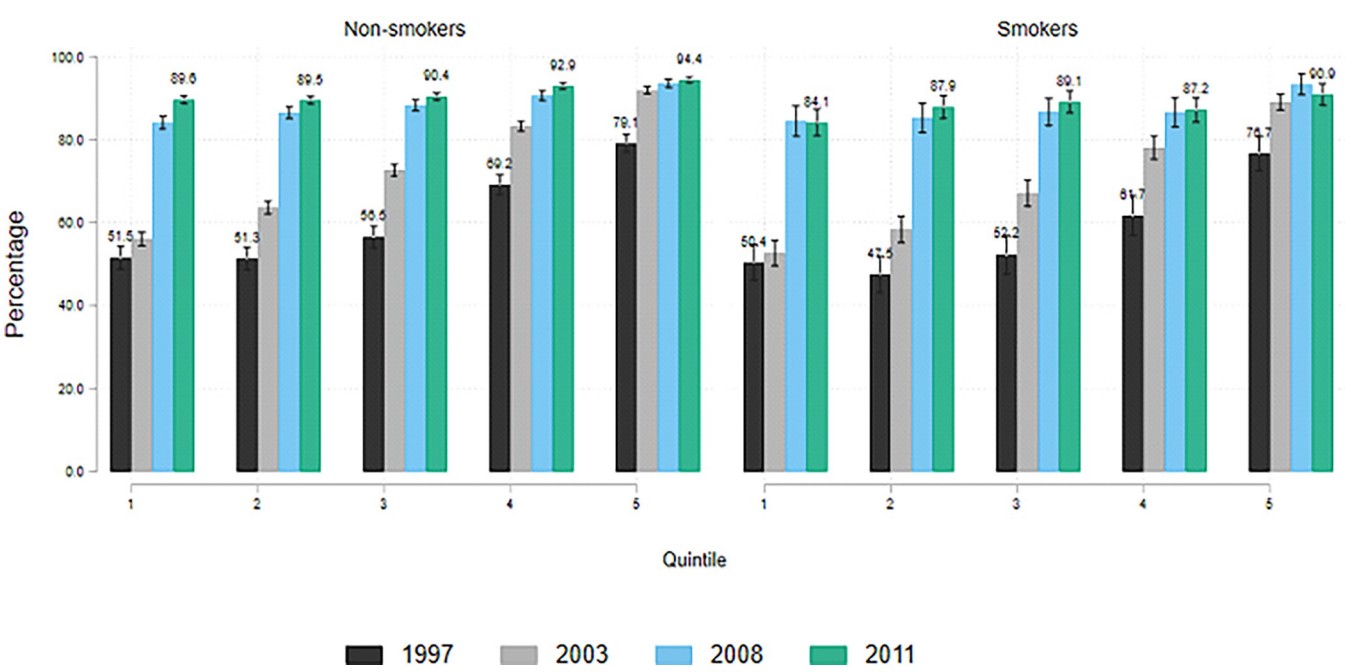

**Fig 3. Self-reported health status and affiliation with the health system.** Label: Authors' calculations.

The universal health insurance policy is reflected in people's health status, whereby in 1997, 60% of respondents in quintile 1 reported that their health was bad, whereas by 2011 this figure had fallen to 37%, while in quintile 5, the proportion was close to 25% in both 1997 and 2011. These substantial improvements in quintile 1 are irrespective of smoking status, which might be related to the fact that the insurance premiums and co-payments are tied to earnings and not to specific risk variables. Besides, individuals are free to move between insurers, limiting the ability of insurers to cream-skim according to risk.

## 3. Methods

Below we present the analysis required for this study. First, we describe the data and the matching strategy, and then the regression analysis. Ethical approval for this type of study is not required by our institute given the use of secondary data.

### 3.1 Data and matching

To obtain tobacco consumption data that reflect changes through time, we used household consumption data that were collected as a part of the ECV for 1997, 2003, 2008, and 2011. These surveys include detailed household consumption records for the previous seven days. Using this information, we constructed monthly equivalent household expenditures using the OECD method. As described above, these data allow us to determine (i) whether there is a tobacco user in the household (prevalence, based on expenditures) and (ii) the share of budget expenditure on the following categories: tobacco, food, alcohol, clothing, household services, health, education, transport, and other items. Online appendix A describes how these categories were constructed. The shares are calculated based on total expenditure including tobacco.

One important concern when comparing households of smokers across time is the composition differences: apart from time, the notorious reduction in smoking prevalence is not random. Thus, different budget shares might be the result of different needs of the households. Our goal with the matching is to replicate the characteristics of smokers of 2011 with those of smokers and non-smokers from the previous years. For this, we implement a genetic matching algorithm which has been used before for assessing crowding-out effects from tobacco [24]. The method uses genetic optimization to choose a group of $M$ control units per treatment unit, which are as closely as possible in a vector of characteristics [43]. The method chooses the metric that is used to measure the distance between the vectors, where the objective is to minimise the bias between treatment and the conformed comparison group (i.e. maximise the balance). Typically, the propensity score matching is added as an additional covariate. In our implementation, we search for one smoker, and one non-smoker, per each 2011 smoker household–treatment group—($M$ = 1), each year, in each expenditure quintile. The method was implemented with the package *Matching* in R [44]. We consider as a robustness check a kernel propensity score matching.

Table 1 compares non-smoker households and smoker households per year. For each variable -all of them considered in the matching algorithm -, we observe the mean for both groups per year. In the first row before matching and in the second after it. The asterisks reflect the level of significance of a comparison of the means of smokers with non-smokers. The goal of matching is to ensure similar distributions of the covariates, not only that the mean of each covariate is the same.

Overall, in all years, we observe that relative to earlier cohorts, 2011 households that bought cigarettes had household heads who were older, more likely to be female, more educated, and lived in smaller households with fewer children. These trends also apply to the non-smoker population. The matching strategy reduces those differences, but as differences are not completely gone, the econometric model below includes these variables as controls. Therefore, our study is based on the expenditure's composition for smoker households similar to the ones observed in 2011, which change over time.

Appendix Table B.2 in S1 File shows the balance after the kernel matching, which also reduces substantially differences. Still, overall it is less successful than the genetic matching, mostly coming from 2003 non-smoker population.

**Table 1. Matching sample balance.**

| Variable | Sample | 1997 | | 2003 | | 2008 | | 2011 | |
|---|---|---|---|---|---|---|---|---|---|
| | | Smoker | Non-Smoker | Smoker | Non-Smoker | Smoker | Non-Smoker | Smoker | Non-Smoker |
| Age | NM | 47.690*** | 46.467*** | 46.616*** | 46.927*** | 49.271 | 47.342*** | 49.939 | 48.001*** |
| | M | 49.241 | 48.363 | 48.613 | 47.267 | 49.946 | 49.094 | | 48.705 |
| Gender (Female = 1) | NM | 0.205*** | 0.258 | 0.272 | 0.328*** | 0.228*** | 0.323*** | 0.270 | 0.323*** |
| | M | 0.229** | 0.222 | 0.225 | 0.206 | 0.257 | 0.220 | | 0.191 |
| Primary school | NM | 0.842** | 0.793 | 0.634*** | 0.616*** | 0.830** | 0.752*** | 0.811 | 0.700*** |
| | M | 0.824 | 0.823 | 0.781 | 0.742 | 0.817 | 0.817 | | 0.812 |
| Secondary school | NM | 0.077 | 0.100 | 0.130*** | 0.154*** | 0.093 | 0.147*** | 0.089 | 0.153*** |
| | M | 0.090 | 0.079 | 0.077 | 0.078 | 0.083 | 0.077 | | 0.067 |
| Tertiary school | NM | 0.081* | 0.107 | 0.236*** | 0.230*** | 0.078*** | 0.101 | 0.099 | 0.147*** |
| | M | 0.086 | 0.098 | 0.142 | 0.180 | 0.101 | 0.106 | | 0.121 |
| Zone (Urban = 1) | NM | 0.540 | 0.609*** | 0.773*** | 0.814*** | 0.537 | 0.604*** | 0.539 | 0.584*** |
| | M | 0.531 | 0.516 | 0.560 | 0.625 | 0.523 | 0.516 | | 0.487 |
| Ratio children-under-5/adults | NM | 0.801*** | 0.782*** | 0.580*** | 0.637*** | 0.590*** | 0.685*** | 0.515 | 0.652*** |
| | M | 0.516 | 0.553 | 0.478** | 0.510 | 0.501 | 0.500 | | 0.492 |
| Total individuals | NM | 4.612*** | 4.106*** | 3.922 | 3.665*** | 4.226*** | 3.978 | 3.919 | 3.876 |
| | M | 3.750** | 3.777 | 3.753 | 3.543 | 3.825 | 3.620 | | 3.477 |

*Notes*: Per variable, the first row corresponds to the sample without matching (NM), and the second to the matched sample (M). Genetic matching with the propensity score, with five neighbours, population size of the optimizer of 10000. Significance of t-test between smokers of each year, and smokers of 2011

* 10%

** 5%

*** 1%.

## 3.2 Empirical strategy

Our goal is to determine differences overtime (1997 to 2011) on how budget-pressure of smoking affects the poorest households, relative to the richest ones. To compare conditional means, we use a linear model over a comparable sample of individuals. As exposed above, with our matching strategy, we ensure that the observed characteristics, which determine household expenditures, are comparable.

Here we compare, in a cross-section analysis, budget shares between smokers and non-smokers. For each year, we estimate the regression

$$b_i^{(j)} = \alpha_0^{(j,s)} \cdot s_i + \alpha_0^{(j,ns)} \cdot ns_i + \sum_{l=2}^{5} [\alpha_l^{(j,s)} Q_{it}^{(l)} \cdot s_i + \alpha_l^{(j,ns)} Q_{it}^{(l)} \cdot ns_i] + \gamma^{(j,k)} X_i + e_i^{(j)} \tag{1}$$

where $s_i$ and $ns_i$ are dummy variables indicating whether the household has a smoker. The vector $X_i$ represents the control variables, which are log-expenditures, squared log-expenditures, log-age, female dummy, education level dummies, a dummy for living in an urban area, the ratio of household members under 5 per adult, household size, and log-income. Then, the parameter $\alpha_l^{(j,s)}$ presents, for smokers, the difference between budget-share for item $j$ of households in quintile $l$ relative to quintile 1 (lowest quintile of the total expenditure adjusted for household composition, 'the *poorest*'). For non-smokers, the parameter $\alpha_l^{(j,ns)}$ does the same. Crowding-in/out for quintile 1 can be tested with the null $\alpha_0^{(j,s)} = \alpha_0^{(j,ns)}$. Whether the smoking status of the household is relevant for budget-share inequalities can tested with the null $\alpha_l^{(j,s)} = \alpha_l^{(j,ns)}$.

For each year, we estimate a Seemingly Unrelated Regression model (SUR), in which unobserved terms $e_i^{(j)}$ are correlated across spending categories, since households simultaneously decide the proportion of income spent in each good group and are constrained by a single budget constraint.

## 4. Results

Rows A of Table 2 shows the average budget share for each expenditure category after matching for smokers in expenditure quintile 1 (the *poorest*). Rows C do the same for non-smokers. Some expenditure categories, such as transport, grew over time, while there were reductions in expenditure on health, education, and clothing. This pattern is likely to be the result of substantial reductions in the costs of health and education services due to the roll-out of social policies. During the study period, full coverage was achieved in relation to health insurance and basic education, mainly because of efforts in the public sector.

What we are interested in is the difference in trends between smokers and non-smokers over time. Rows E in the table present the p-vale of a Wald test between the budget shares per item is the same for smokers and non-smokers in the poorest quintile. Smokers' households tend to spend more on alcohol and less on transport and housing most years. There is no evidence of crowding-out in health (negative but non-significant coefficients, but there is for education in 1997 and 2011. For food, smokers' households devoted fewer resources in 2008, it is also negative for 1997 and 2011, but not significant at the 90% level.

Next, we consider how different households of the fifth quintile (the *richest*) with those of the first (the *poorest*) in terms of budget shares. Rows B (smokers) and D (non-smokers) present such differences. As usual, richer households devote a smaller proportion to food consumption, and more to clothing and other expenditures. However, how different are those gradients between smokers' and non-smokers' households? Rows F test how different are those gradients. First, the alcohol crowding-in seems larger for richer households only in 2003. Second, we observe that for richer households the observed smaller share of expenditures for food of smokers' households occurs in a smaller magnitude than for non-smokers; for "others", the extra share is small for smokers. Third, there are no significant differences across smokers and non-smokers, for the gap in the shares for health and education between quintiles 5 and 1 (the *richest* to the *poorest*).

As a robustness check, we performed two exercises. First, the plain analysis without matching (Appendix Table B.1 in S1 File). Second, the same analysis but using weights coming from the kernel matching (Appendix Table B.3. in S1 File). The analysis without matching resulted in lower standard errors, and as a result, in more rejected nulls. Still, magnitudes and directions are largely unchanged. In the case of kernel matching, results are closer to those of genetic matching. Hence, we claim that the central messages are qualitatively the same.

## 5. Discussion

As shown in previous studies, financial pressure due to tax increases may affect human capital accumulation [21–23]. The objective of this article is to determine if this is the case for low-SES households, during a period when tobacco framework policies were introduced and at the same time, publicly provided health and education services were expanded.

Crowding-out of human capital accumulation among low SES households might be a possible undesired effect of the tobacco control policies. Price increases might have induced a compositional change in smokers, as occasional consumers are more likely to cease consuming tobacco in response to tax hikes than frequent smokers are. Thus, a larger percentage of households that continue to consume tobacco under a higher-price regime will be composed of

**Table 2. SUR Estimates for variation on shares between smokers and non-smokers.**

| Variable | Alcohol | | | | Others | | | |
|---|---|---|---|---|---|---|---|---|
| | **1997** | **2003** | **2008** | **2011** | **1997** | **2003** | **2008** | **2011** |
| A: Q1 Smokers Share | 0.012 | 0.009 | 0.008 | 0.016 | 0.087 | 0.071 | 0.091 | 0.092 |
| | (0.002) | (0.001) | (0.001) | (0.002) | (0.004) | (0.004) | (0.004) | (0.003) |
| B: Q5 vs Q1 smokers share: $Q_5^{(l)} \cdot s_{it}$ | 0.023 | 0.017 | 0.017 | 0.021 | 0.079 | 0.039 | 0.156 | 0.175 |
| | (0.006) | (0.007) | (0.006) | (0.006) | (0.018) | (0.017) | (0.019) | (0.019) |
| C: Q1 Non-smokers Share | 0.006 | 0.006 | 0.002 | 0.005 | 0.078 | 0.078 | 0.101 | 0.098 |
| | (0.002) | (0.001) | (0.001) | (0.002) | (0.004) | (0.004) | (0.004) | (0.003) |
| D: Q5 vs Q1 non-smokers share: $Q_5^{(l)} \cdot ns_{it}$ | 0.017 | 0.005 | 0.015 | 0.021 | 0.137 | 0.076 | 0.167 | 0.138 |
| | (0.006) | (0.007) | (0.005) | (0.006) | (0.019) | (0.017) | (0.020) | (0.019) |
| E: Share difference smok. vs non-smok. Q1 | 0.006 | 0.003 | 0.006 | 0.012 | 0.007 | -0.006 | -0.013 | -0.009 |
| p-val | 0.010 | 0.177 | 0.002 | 0.000 | 0.233 | 0.328 | 0.044 | 0.124 |
| F: Gradient difference smok. vs non-smok. Q1 | 0.005 | 0.012 | 0.002 | -0.000 | -0.058 | -0.037 | -0.011 | 0.037 |
| p-val | 0.213 | 0.001 | 0.550 | 0.982 | 0.000 | 0.002 | 0.444 | 0.009 |
| **Variable** | **Transport** | | | | **Housing** | | | |
| | 1997 | 2003 | 2008 | 2011 | 1997 | 2003 | 2008 | 2011 |
| A: Q1 Smokers Share | 0.031 | 0.041 | 0.039 | 0.054 | 0.160 | 0.252 | 0.293 | 0.203 |
| | (0.002) | (0.003) | (0.003) | (0.002) | (0.007) | (0.006) | (0.007) | (0.006) |
| B: Q5 vs Q1 smokers share: $Q_5^{(l)} \cdot s_{it}$ | 0.000 | 0.000 | 0.025 | 0.019 | 0.045 | -0.015 | 0.067 | -0.033 |
| | (0.007) | (0.009) | (0.008) | (0.008) | (0.020) | (0.018) | (0.018) | (0.018) |
| C: Q1 Non-smokers Share | 0.034 | 0.051 | 0.051 | 0.050 | 0.190 | 0.282 | 0.298 | 0.247 |
| | (0.002) | (0.003) | (0.003) | (0.002) | (0.007) | (0.006) | (0.007) | (0.006) |
| D: Q5 vs Q1 non-smokers share: $Q_5^{(l)} \cdot ns_{it}$ | -0.003 | -0.013 | 0.016 | 0.027 | 0.011 | -0.012 | 0.069 | -0.060 |
| | (0.007) | (0.009) | (0.008) | (0.008) | (0.020) | (0.017) | (0.017) | (0.018) |
| E: Share difference smok. vs non-smok. Q1 | -0.002 | -0.010 | -0.013 | 0.005 | -0.035 | -0.028 | 0.008 | -0.045 |
| p-val | 0.649 | 0.016 | 0.000 | 0.179 | 0.000 | 0.000 | 0.356 | 0.000 |
| F: Gradient difference smok. vs non-smok. Q1 | 0.002 | 0.012 | 0.009 | -0.008 | 0.034 | -0.003 | -0.002 | 0.027 |
| p-val | 0.647 | 0.042 | 0.074 | 0.161 | 0.016 | 0.824 | 0.852 | 0.059 |
| **Variable** | **Food** | | | | **Clothing** | | | |
| | 1997 | 2003 | 2008 | 2011 | 1997 | 2003 | 2008 | 2011 |
| A: Q1 Smokers Share | 0.502 | 0.455 | 0.474 | 0.526 | 0.131 | 0.106 | 0.053 | 0.057 |
| | (0.008) | (0.006) | (0.008) | (0.007) | (0.009) | (0.006) | (0.004) | (0.003) |
| B: Q5 vs Q1 smokers share: $Q_5^{(l)} \cdot s_{it}$ | -0.099 | -0.045 | -0.294 | -0.205 | 0.019 | 0.017 | 0.037 | 0.030 |
| | (0.019) | (0.018) | (0.022) | (0.020) | (0.021) | (0.014) | (0.012) | (0.007) |
| C: Q1 Non-smokers Share | 0.518 | 0.445 | 0.486 | 0.548 | 0.113 | 0.095 | 0.065 | 0.059 |
| | (0.008) | (0.006) | (0.008) | (0.007) | (0.009) | (0.005) | (0.004) | (0.004) |
| D: Q5 vs Q1 non-smokers share: $Q_5^{(l)} \cdot ns_{it}$ | -0.128 | -0.071 | -0.321 | -0.209 | 0.023 | 0.025 | 0.025 | 0.025 |
| | (0.018) | (0.018) | (0.023) | (0.021) | (0.018) | (0.013) | (0.010) | (0.009) |
| E: Share difference smok. vs non-smok. Q1 | -0.015 | 0.008 | -0.023 | -0.018 | 0.016 | 0.009 | -0.009 | -0.002 |
| p-val | 0.134 | 0.379 | 0.049 | 0.118 | 0.212 | 0.316 | 0.138 | 0.721 |
| F: Gradient difference smok. vs non-smok. Q1 | 0.029 | 0.025 | 0.027 | 0.004 | -0.004 | -0.008 | 0.012 | 0.005 |
| p-val | 0.018 | 0.052 | 0.062 | 0.792 | 0.748 | 0.401 | 0.072 | 0.471 |
| **Variable** | **Health** | | | | **Education** | | | |
| | 1997 | 2003 | 2008 | 2011 | 1997 | 2003 | 2008 | 2011 |
| A: Q1 Smokers Share | 0.050 | 0.030 | 0.027 | 0.034 | 0.027 | 0.033 | 0.001 | 0.000 |
| | (0.003) | (0.002) | (0.003) | (0.002) | (0.003) | (0.002) | (0.001) | (0.001) |
| B: Q5 vs Q1 smokers share: $Q_5^{(l)} \cdot ns_{it}$ | -0.024 | -0.004 | 0.008 | 0.010 | 0.013 | -0.012 | 0.002 | 0.005 |

*(Continued)*

**Table 2.** (Continued)

| Variable | Alcohol | | | | Others | | | |
|---|---|---|---|---|---|---|---|---|
| | 1997 | 2003 | 2008 | 2011 | 1997 | 2003 | 2008 | 2011 |
| | (0.011) | (0.008) | (0.009) | (0.009) | (0.008) | (0.008) | (0.002) | (0.002) |
| C: Q1 Non-smokers Share | 0.054 | 0.035 | 0.033 | 0.034 | 0.037 | 0.032 | 0.001 | 0.002 |
| | (0.003) | (0.002) | (0.003) | (0.002) | (0.003) | (0.002) | (0.001) | (0.001) |
| D: Q5 vs Q1 non-smokers share: $Q_5^{(l)} \cdot ns_{it}$ | -0.003 | -0.010 | 0.009 | 0.026 | -0.011 | -0.013 | 0.004 | 0.005 |
| | (0.012) | (0.010) | (0.010) | (0.010) | (0.008) | (0.007) | (0.002) | (0.003) |
| E: Share difference smok. vs non-smok. Q1 | -0.003 | -0.006 | -0.005 | -0.002 | -0.008 | 0.000 | 0.000 | -0.002 |
| p-val | 0.466 | 0.137 | 0.317 | 0.705 | 0.026 | 0.969 | 0.746 | 0.047 |
| F: Gradient difference smok. vs non-smok. Q1 | -0.021 | 0.006 | -0.001 | -0.015 | 0.024 | 0.000 | -0.002 | -0.001 |
| p-val | 0.006 | 0.338 | 0.916 | 0.018 | 0.000 | 0.980 | 0.139 | 0.680 |

*Notes*: This table summarises the main results with total expenditure net of expenditure on tobacco. Estimates are produced after estimating a SUR on the sample resulting from the genetic matching using ECV 1997, 2003, 2008, and 2011 data. Each set of columns corresponds to a category of spending per year. Quantiles are based on total annual household expenditures adjusted for household composition (Hagennars et al., 1994). In each year, the unconditional shares for smokers and non-smokers from quintile 1 are presented (rows A and C), as well as the difference of these shares for quintile 5 which correspond to Eq 1 estimated coefficients conditional on controls (rows B and D). Below them, two tests compare the previous numbers between smokers and non-smokers (A—C, B—D), both of them computed with the estimates of Eq 1. Controls include log-expenditures, squared log-expenditures, log-age, female dummy, education level dummies (primary or less [base], secondary, tertiary), a dummy that indicates if the household resides in an urban area, the ratio of the number of children under 5 per adult, household size, and log annual-income adjusted for household composition. Standard errors in parentheses.

frequent/heavy consumers [3–6,45]. As a result, the remaining smokers are less sensitive to price and would be more likely to substitute other goods to maintain their habit [13–15]. This crowding-out effect was found in several countries [20–23,32,46].

Our study complements the literature that explores the role of tobacco usage on household expenditures. The main strength of this study is that we can explore crowding-out in a setting of expansion of the welfare state. Most of the literature is based on the comparison of budget shares between smokers and non-smokers in a static context [22]. While studies like Block and Webb (2009), San & Chaloupka (2016), and Mugosa et al (2023) consider several years of information, the treatment of information is still static [23,32,47]. Nyagwachi, Chelwa, and van Walbeek (2020) use a dynamic setting for identification but still, their objective is to measure the amount of crowding out in a given moment [48]. We contribute by showing that crowding out should not be a concern in middle-income countries once a strong welfare state is in place.

In the Colombian case, between 1997 and 2011 there was a notorious increase in the budget share allocated to smoking in comparable low-SES households of smokers and a reduction in the budget share allocated to health and education. Therefore, while financial pressure on smokers was growing via taxes, their disposable income was growing due to the income effect of the social policies. The results presented above show almost no evidence of crowding-out in health and food expenditures, on quintiles neither 1 nor 5. We also observe crowding-in of alcohol most years, which is typically associated with tobacco consumption in the literature. Such differences were not present for highest income households, aside from a larger crowding-in for alcohol. The only difference is that the extra share on clothing and other expenditures of the highest quintile is smaller for smokers than for non-smokers. Therefore, the growth in tobacco expenditures is mostly affecting the leisure, entertainment, and luxury expenses of households.

It is important to mention that this study compares budget shares over time for a 'specific' group: those people who declared to be smokers and who had characteristics similar to those of smokers in 2011. Hence, our estimates do not identify the causal effect of the expansion of social welfare (on health and education) and/or tobacco control policies on the budget shares of smokers. This limitation for the interpretation comes from the potential impacts of those policies on the composition of smokers which our methodology cannot capture.

Finally, our study cannot isolate the effects of the social policy expansion from the changes in tobacco control policies as they occurred simultaneously during the period. Still, as the tobacco control policies were strengthening over time (stronger financial pressure on remaining smokers), ours became a lower bound of the changes that would be derived solely from the expansion of the benefits.

## 6. Conclusions

The tobacco control literature has shown that one of the implications of tobacco consumption is that it results in less disposable income for households. As a result, household reallocate their resources in a way that hampers human capital accumulation. Still, this argument implies that increasing taxes would improve the situation of those quitting smoking but exacerbate the problem of those who remain smoking. Hence, these policies could lead the 'victims' of tobacco control, and their descendants, to poverty. We show that the Government can protect them with effective general policies directed towards health and education, irrespective of the smoking status of households.

The finding above is central for middle-income countries which would want to take the appropriate measures of the WHO FCTC, but where parliaments and governments hesitate based on the impact on their citizens. In the case of Colombia, tobacco prices are still some of the lowest on the continent despite recent tax hikes in 2017, and further efforts are required [12,49]. In general, countries might contain adverse effects on household budgets due to taxing temptation goods when they expand their social security programs.

## Supporting information

**S1 File. The file "S1 File.pdf" includes.**

i. Appendix A. Data

ii. Appendix B. Robustness.
   (PDF)

## Acknowledgments

We thank Geoff Whyte, MBA, from Edanz Group (www.edanzediting.com/ac) for editing a draft of this manuscript. We acknowledge the valuable research assistance by Susana Otálvaro.

## Author Contributions

**Conceptualization:** Guillermo Paraje, Paul Rodríguez-Lesmes.

**Data curation:** Paul Rodríguez-Lesmes.

**Formal analysis:** Juan Miguel Gallego, Guillermo Paraje, Paul Rodríguez-Lesmes.

**Funding acquisition:** Juan Miguel Gallego, Guillermo Paraje.

**Investigation:** Juan Miguel Gallego, Guillermo Paraje, Paul Rodríguez-Lesmes.

**Methodology:** Juan Miguel Gallego, Guillermo Paraje, Paul Rodríguez-Lesmes.

**Project administration:** Juan Miguel Gallego, Guillermo Paraje, Paul Rodríguez-Lesmes.

**Resources:** Paul Rodríguez-Lesmes.

**Software:** Paul Rodríguez-Lesmes.

**Supervision:** Paul Rodríguez-Lesmes.

**Validation:** Juan Miguel Gallego, Guillermo Paraje, Paul Rodríguez-Lesmes.

**Visualization:** Paul Rodríguez-Lesmes.

**Writing – original draft:** Juan Miguel Gallego, Guillermo Paraje, Paul Rodríguez-Lesmes.

**Writing – review & editing:** Juan Miguel Gallego, Guillermo Paraje, Paul Rodríguez-Lesmes.

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
