## [Decision Letter · Decision Letter 0]

11 Mar 2024

PONE-D-23-33110Inequality of the crowding-out effect of tobacco expenditure in ColombiaPLOS ONE

Dear Dr. Rodríguez-Lesmes,

Thank you for submitting your manuscript to PLOS ONE. After careful consideration, we feel that it has merit but does not fully meet PLOS ONE’s publication criteria as it currently stands. Therefore, we invite you to submit a revised version of the manuscript that addresses the points raised during the review process.

The research paper, titled "Inequality of the Crowding-Out Effect of Tobacco Expenditure in Colombia," delves into the impact of tobacco tax on the budget allocations of low socioeconomic status (SES) households. Through the application of genetic matching methods, the study establishes a counterfactual group for smoking households. The primary aim is to examine the differences in consumption shares between smoking households and the control group, seeking to determine whether smoking behavior results in a crowding-out effect on other expenditures.

 While the methodology employed in the article is relatively innovative, a universally accepted and robust testing procedure is currently lacking. Nevertheless, the technical process appears sound without discernible flaws.

Despite these strengths, the paper faces several issues that require attention for publication.  Key concerns include:

1) The data utilized spans a relatively extended period, and discussions about this timeframe should consider the potential impact of exogenous shocks.

2) The reliance on genetic matching prompts consideration of whether incorporating other matching methods could enhance result robustness.

3) The empirical conclusions section is criticized for being a mere enumeration of results, lacking in-depth discussion.

4) It is essential to clarify whether the first quintile to the fifth quintile represents the poorest to the richest or vice versa.

5) What are the policy implications based on the findings?

6) Additionally, what is the contribution, innovation, and limitation of this study?==============================

We look forward to receiving your revised manuscript.

Kind regards,

Enrique Teran

Academic Editor

PLOS ONE

Journal Requirements:

4. Please be informed that funding information should not appear in the Acknowledgments section or other areas of your manuscript. We will only publish funding information present in the Funding Statement section of the online submission form. Please remove any funding-related text from the manuscript.

Reviewers' comments:

Reviewer's Responses to Questions

**Comments to the Author**

1. Is the manuscript technically sound, and do the data support the conclusions?

Reviewer #1: Yes

2. Has the statistical analysis been performed appropriately and rigorously? 

Reviewer #1: Yes

3. Have the authors made all data underlying the findings in their manuscript fully available?

Reviewer #1: Yes

4. Is the manuscript presented in an intelligible fashion and written in standard English?

Reviewer #1: Yes

5. Review Comments to the Author

Reviewer #1: The research paper, titled "Inequality of the Crowding-Out Effect of Tobacco Expenditure in Colombia," delves into the impact of tobacco tax on the budget allocations of low socioeconomic status (SES) households. Through the application of genetic matching methods, the study establishes a counterfactual group for smoking households. The primary aim is to examine the differences in consumption shares between smoking households and the control group, seeking to determine whether smoking behavior results in a crowding-out effect on other expenditures.

While the methodology employed in the article is relatively innovative, a universally accepted and robust testing procedure is currently lacking. Nevertheless, the technical process appears sound without discernible flaws.

Despite these strengths, the paper faces several issues that require attention for publication. Key concerns include:

1) The data utilized spans a relatively extended period, and discussions about this timeframe should consider the potential impact of exogenous shocks.

2) The reliance on genetic matching prompts consideration of whether incorporating other matching methods could enhance result robustness.

3) The empirical conclusions section is criticized for being a mere enumeration of results, lacking in-depth discussion.

4) It is essential to clarify whether the first quintile to the fifth quintile represents the poorest to the richest or vice versa.

5) What are the policy implications based on the findings?

6) Additionally, what is the contribution, innovation, and limitation of this study?

Addressing these concerns comprehensively will significantly fortify the paper and improve its suitability for publication.

6. PLOS authors have the option to publish the peer review history of their article (what does this mean?). If published, this will include your full peer review and any attached files.

Reviewer #1: **Yes: **Rose Zheng

---

## [Author Response · Author response to Decision Letter 0]

17 Apr 2024

We have carefully considered each of the reviewers' comments and have made the following revisions to address their concerns. We slightly changed the order of the comments and group two of them together.

3) The empirical conclusions section is criticized for being a mere enumeration of results, lacking in-depth discussion.

6) Additionally, what is the contribution, innovation, and limitation of this study?

Addressing these concerns comprehensively will significantly fortify the paper and improve its suitability for publication.

R: Many thanks for these recommendations, certainly we were not highlighting enough such elements. We have rewritten the discussion and conclusion sections so they reflect the central elements:

i. Our central innovation is on studying crowding-out in a dynamic setting, specifically, during the expansion of the Colombian welfare state. This is different from the existing literature as most papers concentrate on measuring crowding-out at a given point in time. It also means that our ‘methods’ differ from the standard studies given the different objectives.

ii. Our contribution is to show that crowding out of expenditures linked to human capital accumulation is less relevant once a strong welfare state is in place.

iii. The central limitation is that while the matching allows us to compare a relatively homogenous group across time, we cannot estimate the ‘impact’ of the social policy expansions on budget shares as we do not have an adequate control group of ‘treated’ smokers vs ’non-treated’ smokers. In other words, we are estimating an object closer to a local average treatment effect (LATE) rather than an average treatment effect on smokers (ATT). Here the ‘local’ group corresponds to smokers that had characteristics similar to those of smokers in 2011, rather than the ‘compliers’ of an instrumental variables’ setup.

iv. Another limitation is that we cannot isolate the effects of the social policy expansion from the changes in tobacco control policies as they occur simultaneously during the period. Still, as the tobacco control policies were strengthening over time, ours became a lower bound of the changes that would be derived solely by the expansion of the benefits.

1) The data utilized spans a relatively extended period, and discussions about this timeframe should consider the potential impact of exogenous shocks.

R: Indeed, it is a relatively extended period. Such period was selected based on (i) availability of data, (ii) tobacco control interventions, and (iii) welfare policies. During this period many exogenous shocks have occurred. Still, as we do a comparison within year of smokers vs. non-smokers across expenditure quantiles, specific time shocks are removed. Yet, as the strategy is still challenged by the composition of who are the smokers (more a preferences trend than a year-specific shock), the matching strategy aims to establish a group as homogenous as possible to minimize such concern.

The central limitation here is our inability to disentangle the change in tobacco control policies from the changes in the welfare state. We now acknowledge this limitation in the discussion as described above.

2) The reliance on genetic matching prompts consideration of whether incorporating other matching methods could enhance result robustness.

R: In principle, we did not include results with other methods as the genetic matching is one of the best options available. But we see the point on showing that the results are not an artifact of the method. For this reason, as a supplemental material (B. Robustness), we have included results without matching, and with a traditional propensity score kernel matching. In the main text, we included a paragraph at the end of the results section discussing these alternatives. In general, results are qualitatively the same. 

4) It is essential to clarify whether the first quintile to the fifth quintile represents the poorest to the richest or vice versa.

R: Many thanks for pointing this out. We have made clear in the methods, in the results, and in the Table notes that we refer to quintiles of total expenditure over equivalised household size. Hence, quintile 1 corresponds to the ‘poorest’, and quintile 5 to the ‘richest’. In Table 2, the difference is from quintile 5 to quintile 1 (the richest to the poorest).

5) What are the policy implications based on the findings?

Our message is simple: Governments and parliaments of middle-income countries should not refrain from increasing taxes due to crowding out of expenditures.

R: In the tobacco control literature, the are several studies that show that one of the implications of tobacco consumption is that it results in less disposable income for households. Still, this argument implies that increasing taxes would improve the situation of those quitting smoking but exacerbate the problem of those who remain smoking. Hence, these policies could lead the ‘victims’ of tobacco control, and their descendants, to poverty. We show that the Government can protect them with effective general policies directed towards health and education, irrespective of the smoking status of households.

We have rewritten the conclusions to make this point salient.

We believe that these revisions have strengthened the manuscript and have addressed the concerns raised by the reviewers comprehensively.

---

## [Editor Report · Decision Letter 1]

24 Apr 2024

Inequality of the crowding-out effect of tobacco expenditure in Colombia

PONE-D-23-33110R1

Dear Dr. Rodríguez-Lesmes,

We’re pleased to inform you that your manuscript has been judged scientifically suitable for publication and will be formally accepted for publication once it meets all outstanding technical requirements.

Kind regards,

Enrique Teran

Academic Editor

PLOS ONE

Additional Editor Comments (optional):

Thank you for considered the points made by reviewers and get back with a more robust manuscript. It will be now suitable for publication. Congratulations!
---

## [Editor Report · Acceptance letter]

3 May 2024

PONE-D-23-33110R1 

PLOS ONE

Dear Dr. Rodríguez-Lesmes, 

I'm pleased to inform you that your manuscript has been deemed suitable for publication in PLOS ONE. Congratulations! Your manuscript is now being handed over to our production team.

Kind regards, 

on behalf of

Dr. Enrique Teran 

Academic Editor

PLOS ONE